# The Development of a Polysaccharide-Based Hydrogel Encapsulating Tobramycin-Loaded Gelatine Microspheres as an Antibacterial System

**DOI:** 10.3390/gels9030219

**Published:** 2023-03-14

**Authors:** Mingsheng Shi, Yongmeng Xu, Shuai Li, Lifeng Wang, Junyao Gu, Yi-Xuan Zhang

**Affiliations:** 1School of Life Science and Biopharmaceutics, Shenyang Pharmaceutical University, Shenyang 110016, China; 2Woundhealing (Hangzhou) Biotechnology Co., Ltd., Hangzhou 310018, China; 3Shenyang Yaoda Leiyunshang Pharmaceutical Co., Ltd., Benxi 114004, China

**Keywords:** wound infection, quaternary ammonium salts, chitosan, alginate, tobramycin

## Abstract

Bacterial infection contributes to the bioburden of wounds, which is an essential factor in determining whether a wound can heal. Wound dressings with antibacterial properties that can promote wound-healing are highly desired for the treatment of chronic wound infections. Herein, we fabricated a simple polysaccharide-based hydrogel dressing encapsulating tobramycin-loaded gelatine microspheres with good antibacterial activity and biocompatibility. We first synthesised long-chain quaternary ammonium salts (QAS) by the reaction of tertiary amines with epichlorohydrin. The amino groups of carboxymethyl chitosan were then conjugated with QAS through the ring-opening reaction and QAS-modified chitosan (CMCS) was obtained. The antibacterial analysis showed that both QAS and CMCS could kill E. coli and S. aureus at relatively low concentrations. QAS with 16 carbon atoms has a MIC of 16 μg/mL for *E. coli* and 2 μg/mL for *S. aureus*. A series of formulations of tobramycin-loaded gelatine microspheres (TOB-G) were generated and the best formulation was selected by comparing the characters of the microspheres. The microsphere fabricated by 0.1 mL GTA was selected as the optimal candidate. We then used CMCS, TOB-G, and sodium alginate (SA) to prepare physically crosslinking hydrogels using CaCl_2_ and investigated the mechanical properties, antibacterial activity, and biocompatibility of the hydrogels. In summary, the hydrogel dressing we produced can be used as an ideal alternative for the management of bacteria-infected wounds.

## 1. Introduction

Microbial infection has become increasingly recognized as one of the major causes of chronic wounds. It has been reported that the medical costs of wound management are approaching 96.8 billion each year, and recent consensus reports call for the early and aggressive treatment of microbes in chronic wounds [1]. Wound infections are common, serious, and expensive complications after injuries and can progress from colonization and local infection to systemic infection, sepsis, and multiple organ dysfunction syndrome, even death [2]. Infection results in nonhealing and it is appropriate to treat the infected wounds with a combination of topical antibacterial and systemic antibiotics, especially in the presence of invasive infections [3,4,5]. The most commonly isolated organisms of microbes vary and include common pathogens such as *Staphylococcus aureus* (*S. aureus*), *Pseudomonas aeruginosa* (*P. aeruginosa*), and *Escherichia coli* (*E. coli*) [6,7]. Systemic or topical antibiotics application has proven efficacy against wound infection. However, the inappropriate use of systemic antibiotic places the patient at risk for the acquisition of drug resistance, and the topical application of antibiotics has many disadvantages, including contact toxicity and safety issues, such as dermatitis reactions, and difficulties in accurately determining the dose. Concerns over the use of antibiotics and the search for new antimicrobial agents have led to the emergence of advanced tissue engineering technology.

Tobramycin (TOB), is a polycationic aminoglycoside antibiotic with abundant amine groups that can treat various types of infections caused by Gram-negative and -positive bacteria [8]. TOB works mainly by inhibiting bacterial protein synthesis and binding to bacterial outer membranes [9]. Although TOB exhibits excellent antibacterial activity, such small molecule drug may diffuse quickly on the surface of the infected wounds and lead to burst release [10]. On the other hand, the misuse or overuse of TOB may also develop drug-resistance. Therefore, the development of a TOB release system is promising for bacterially infected wounds and the correct topical use of TOB. Due to the high solubility and small molecular character of TOB, a particle release system is needed to effectively control TOB release. Gelatine, a denatured protein, is one of the most natural biopolymers widely used in pharmaceutical industries due to its biodegradability, biocompatibility, and ease of chemical modification and crosslinking [11]. These properties encourage gelatine’s use as particle/nanoparticle delivery systems for drugs [11,12]. Glutaraldehyde (GTA) is a relatively widely used reagent for gelatine crosslinking. By optimising the degree of GTA crosslinking, improved properties of TOB-loaded gelatine particles can be obtained at the desired release rate, suitable particle size, proper porosity, appropriate mechanical strength, and less toxicity.

Hydrogel dressing is claimed as one of the most effective alternatives for wound treatment [13]. Hydrogels are three-dimensional, have high-water content (over 90%), and are crosslinked networks with a certain porosity. Hydrogels are fabricated by the physical or chemical crosslinking of natural and/or synthetic polymers and are able to delivery bioactive [14,15,16]. All of these factors make hydrogels interesting candidates for future smart wound-dressing platform. Hydrogel dressing provides a biocompatible platform, cooling effect, adequate moisture, and the ability to isolate bacteria, which play significant roles in promoting wound-healing [13,17]. Hydrogel dressing can also function as a delivery platform, allowing it to be a promising alternative for hard-to-treat wounds, such as infected chronic wounds [18,19]. Therefore, the design and development of a multifunctional hydrogel dressing is emerging as a promising strategy for antibacterial treatment. The advantages of chemical crosslinking are highly stable network structures ensured by covalent bonds between different polymer chains, high mechanical properties, and thermal resistance. However, toxic potentials caused by the residual chemical and organic solvents during chemical crosslinking may release toxicity. Physically crosslinked hydrogels are produced without the use of chemical modification or the addition of crosslinking agents. Compared with chemically crosslinked hydrogels, such networks exhibit weak mechanical properties, as they are more cell-compatible and more environmentally friendly.

Among the most used natural materials, polysaccharides, which are biocompatible and biodegradable materials, have attracted much attention in the application of wound dressings, drug release, and tissue engineering [20]. Polysaccharides are abundant in nature and reproducible. Chitosan, alginate, cellulose, hyaluronic acid, chondroitin sulphate, and starch are commonly used polysaccharides for hydrogel fabrication. Additionally, each of these materials has its own benefits and drawbacks, which are critical features when deciding its biomedical applications. For example, alginate is an anionic polymer extracted from brown algae, such as *Macrocystis pyrifera*, laminaria digitate by treatment with aqueous alkali solutions [21]. Alginate has been extensively investigated and used for may biomedical applications, due to its biocompatibility, low cost, and mild gelation conditions. Alginate hydrogel-based dressing can be formed via a simple and effective approach, namely an ionic crosslinking method using calcium or other multivalent ions [22]. Alginate dressing facilitates cell migration, provides a moist environment, absorbs exudate, and does not adhere to the wound tissue [23]. However, one critical drawback of ionically crosslinked alginate hydrogel is its relatively short-term stability in physiological conditions due to the release of divalent ions into the surrounding environment. Moreover, alginate hydrogel alone as wound dressing is insufficient for solving bacterial infection, especially in chronic wounds. Chitosan is derived from chitin via an alkaline deacetylation technique and has randomly distributed D-glucosamine and N-Acetyl-D-glucosamine units in its backbone [24]. Chitosan is a promising biomaterial for the fabrication of wound dressings due to a number of advantages, including its low cost, good biocompatibility, basic structural unit that mimics a mammalian extracellular matrix (ECM), and ease of processing [25,26]. More importantly, chitosan has abundant positive charges on the main chain, which interact with the bacterial membrane, suggesting its potential benefits as an antibacterial material against bacteria-infected wounds [27,28]. Films, hydrogels, sponges, and membrane powders have been made from chitosan and extensively studied. The major drawbacks of chitosan are its insolubility in water, mechanical insufficiency, and non-resistance to acid, which limit its biomedical applications. Chemical modification has been used to produce chitosan derivatives with controlled solubility, ionic characteristics, and enhanced antibacterial performance. Among the modification approaches, the quaternary ammonium salts (QAS) of chitosan are the most promising for biomedical applications. Quaternisation is an efficient form of modification due to its desirable solubility, its retention of most of the properties of the modified materials (mainly polysaccharide), and its ability to reduce infections [29,30]. QAS as antibacterial constituents can be introduced into the abundant reactive amino and hydroxyl groups of the chitosan skeleton [31]. Moreover, QAS-modified chitosan improves the solubility of chitosan, thereby expanding the application range of chitosan into different biomedical fields [29]. Therefore, a TOB release hydrogel system based on QAS-modified chitosan and alginate have significant potential in the treatment of infected wounds.

The objective of this study is to develop a simple polysaccharide-based hydrogel system with outstanding antibacterial ability through QAS-modified chitosan and the release of TOB-loaded gelatine particles. Firstly, QAS with different carbon chains were synthesised and characterised to confirm the antibacterial activity. QAS-modified chitosan was then synthesised by the ring opening method, where amino groups of chitosan become quaternary amines. GTA-crosslinked gelatine particles as a TOB drug delivery system were formed and optimised GAT formulation and particle profiles were studied. Lastly, alginate, QAS-modified chitosan, and TOB-loaded gelatine particles were mixed together with calcium to form a homogenous hydrogel-based drug release system. The morphology and the antibacterial performance of the combined hydrogel dressings were investigated. Key benefits of TOB-loaded particles include the avoidance of rapid release from hydrogel, prolonged retention at the target site, and reduced risk of serious dose-related side effects. This study will provide a strategy for treating bacteria-infected wounds.

## 2. Results and Discussion

### 2.1. Preparation and Characterisation of QAS

In this study, we aimed to develop a polysaccharide-based antibiotic release hydrogel to manage wounds susceptible to bacterial infection (Figure 1). First, a series of quaternary ammonium salts with various carbon chains were synthesised and their structure and antibacterial effects were identified. 

The chemical shifts δ = 1.24 ppm and δ = 0.87 ppm of ^1^H-NMR are the peaks of quaternary ammonium N; H on the carbon chain and methyl H at the end of the carbon chain (Figure 2A–D). The chemical shift δ = 3.48 ppm is two methyl H, associated with quaternary ammonium N, and the appearance of this peak indicates the successful synthesis of QAS. The carbon chain length can be obtained according to the ratio of integrated peak area. With terminal methyl H as the basic unit, the peak area ratios of carbon chain H to methyl H are 3.83:1, 5.01:1, 5.59:1, and 6.08:1, respectively, which are close to the theoretical values of 4.00:1, 5.33:1, 6.67:1, and 8.00:1. These data indicate that the QAS materials obtained exhibited different carbon chain lengths and can be distinguished by ^1^H-NMR. The peak area ratios of the two methyl H and terminal methyl H associated with quaternised N were 1.91:1, 1.89:1, 1.97:1, and 1.61:1, which were also close to the theoretical value of 2:1.

The epoxy values of C_10_, C_12_, C_14_, and C_16_ QAS were 13.3254%, 15.5526%, 22.2877%, and 15.9615%, respectively (Table 1). The epoxide values were not high, indicating that most of the epoxides of the synthesised QAS had been ring-opened. In addition to further improving the reaction conditions to increase the epoxide value, the cyclic process should be considered in the subsequent reaction.

### 2.2. Antibacterial Activity of the QAS

The antibacterial activity of the quaternary ammonium salts was determined through the testing of different concentrations of the polymers against Gram-positive bacteria (*S. aureus*) and Gram-negative bacteria (*E. coli*) by MIC values. As shown in Figure 2E,F, all of the quaternary ammonium salts with different carbon chain lengths exhibited antimicrobial activity against both bacteria strains and showed increased activity with an increase in the length of the carbon chain. The MIC data presented in Table 2 show that quaternary ammonium salt with 16 carbon atoms (Eq-N_+_-C^16^) exhibited the strongest activity again both *E. coli* (16 μg/mL) and *S. aureus* (2 μg/mL).

### 2.3. Synthesis and Characterisation of Quaternary Ammonium Salt-Modified Chitosan (CMCS)

From the above study, EP-N^+^-C_10_ was found to have the worst antibacterial effect. Therefore, the other three quaternary ammonium salt-modified chitosan derivatives were synthesised and denoted by CMCS-N^+^-C_12_, CMCS-N^+^-C_14_, and CMCS-N^+^-C_16_. By comparing the ^1^H-NMR results of carboxymethyl chitosan (Figure 3A) with CMCS, it can be seen that the chemical shift δ = 1.3 ppm appears on the spectra after the addition of quaternary ammonium salts, this signal is assigned to the CH_2_ of the alkyl chains. (Figure 3B–D). The results showed that the quaternary ammonium salt modified carboxymethyl chitosan was synthesised successfully.

### 2.4. Antibacterial Effect of CMCS

The influence of the CMCS on the antimicrobial properties was evaluated. As shown in Figure 3E,F, all of the modified chitosan exhibited antibacterial effect while no significant differences between each modified chitosan was observed in either bacterium strain. Moreover, the antibacterial efficacy was weaker than QAS. The possible reason for the significant decrease in antibacterial efficiency could be caused by the partial neutralisation of the positive charge moiety in QAS by chitosan. In addition, free QAS are mainly used to alter cell permeability by aggregating on the surface of the bacterium, but when QAS are conjugated to polymer chains, their ability to alter cell permeability is greatly reduced.

Thus, the MIC of modified chitosan was not discussed in this study. Modified chitosan with 16 carbon atoms (Eq-N_+_-C^16^) was selected for the further experiments.

### 2.5. TOB-Loaded Gelatine Microspheres (TOB-G)

Glutaraldehyde (GTA) is one of the commonly used crosslinking agents for collagen- and gelatine-based materials. To optimise the crosslinked gelatine microspheres, a series of GTA amounts were used to crosslink gelatine while encapsulating tobramycin. Figure 4A–E shows the morphology of the prepared gelatine microspheres through different GTA amounts. From the SEM images, microspheres formed by 0.1 mL and 0.15 mL GTA exhibited the best morphology. With a decrease in the GTA amount used for gelatine crosslinking, the microspheres lost their spherical shape and stuck to one another. Higher amounts of GTA resulted in heterogeneous microspheres and adhesion due to excess crosslinking reagents. Microspheres prepared by 0.1 mL GTA exhibited uniform particle sizes and good dispersion in comparison with other groups. The sizes of the microspheres were distributed in the range of 20 ± 70 μm, with an average diameter of approximately 57 μm. Thus, the formulation of 0.1 mL GTA was selected for the fabrication of TOB-loaded gelatine microspheres. Gelatine microspheres without TOB were formed using 0.1 mL GTA as the control group for further study.

The results in Figure 4F show that in the slightly acidic microenvironment, TOB is rapidly released from microspheres and 30% of the total drug is released in 12 h, while only 10% TOB was released in the physiological microenvironment at this time. TOB release reached over 50% at day 6 for the acidic environment and 30% for the physiological microenvironment. These results indicate that the TOB release profile can be adjusted by changing the pH environment, which is due to the pH responsive structure of the crosslink formed by the Schiff base reaction of glutaraldehyde with amino and aldehyde groups in gelatine [15]. The pH sensitivity of hydrogel dressings show potential as a promising material for the treatment of infected wounds. Excessive microbial growth affects the pH of the wound environmental conditions. Dressings with pH sensitivity allows the release of loaded drugs in a more controlled and on-demand manner, providing effective ideas to develop antimicrobial dressings.

### 2.6. Preparation and Characterisation of the TOB-Loaded Polysaccharide Hydrogels (CMCS/SA/TOB-G)

Sodium alginate (SA) and QSA modified chitosan (CMCS) have been proven to form a physically crosslinked polysaccharide hydrogel network through the electrostatic interaction between polyanion and polycation and hydrogen bonding [32]. In this work, we used CaCl_2_ to induce alginate physical crosslinking and balance the charge of alginate and QSA-modified chitosan. TOB-loaded gelatine microspheres were added to QSA-modified chitosan before mixing with alginate solution. QSA-modified chitosan with different carbon chains were used to crosslink with alginate to compare the influences of carbon chains of QSA on the mechanical properties of the hydrogels (Figure 5A,B). The rheological behaviour of the CMCS/SA/TOB-G hydrogels was investigated by an oscillatory rheology and was shown in Figure 5C–E The storage (G′) and loss (G″) modulus represent the mechanical properties. All four hydrogels formed exhibited a similar storage modulus with a range of 700–1070 Pa without any relation in the length of carbon chain on CMCS. The strain sweep curve in the figure confirms the gel-like behaviour, and the hydrogels maintained their strength well under a shear force between 1% and 100%.

Based on the abovementioned results, we selected CMCS-N^+^-C_16_ and 0.1 mL GTA-produced TOB-G as the optimised components for generating the biocompatible and antibacterial hydrogel dressing and further experiments.

### 2.7. Antibacterial Effect of the CMCS/SA/TOB-G Hydrogel

The representative images of bacteria treated with CMCS/SA/TOB-G, TOB-G, and CMCS-N^+^-C_16_ were shown in Figure 6. As expected, CMCS/SA/TOB-G exhibited the strongest antibacterial activity against both *E. coli* and *S. aureus*. The hydrogel killed over 95% of *S. aureus* at a concentration of 128 μg/mL and *E. coli* at 512 μg/mL (Figure 6C–E). TOB-G demonstrated antibacterial activity against both bacteria strains, which is caused by the TOB released from the microspheres. The CMCS group did not show satisfactory antibacterial activity, which is due to the immobilisation of CMCS in the agar plates and the limited contact with bacteria that occurred during the experiment.

### 2.8. Morphology of the CMCS/SA/TOB-G Hydrogel

SEM was performed to assess the hydrogel’s microstructure. The SEM images in Figure 7A show that the scaffold has a highly porous structure and the pore size is about 110 ± 20 μm. This advantage reveals that it is possible to load a high number of drug-loaded microspheres (Figure 7B).

### 2.9. Cytocompatibility

Good biocompatibility is one of the basic requirements of biomedical devices. To investigate the cytocompatibility of the hydrogel combination, we used MTT assay and live/dead staining. The hydrogel extract showed over 90% cell viability after 24 and 48 h of contact (Figure 7C). As shown in Figure 7D, live/dead staining investigations revealed that a majority of cells were alive and remained healthy after co-culturing with the hydrogel extract. The result of live/dead staining was consistent with the MTT result, indicating that the hydrogel has good biocompatibility and can be used as a medical wound dressing.

## 3. Conclusions

In this report, we prepared a simple polysaccharide-based hydrogel system encapsulating TOB-loaded gelatine microspheres as an antibacterial wound dressing. The findings demonstrated that the antibacterial activity of QAS-modified chitosan depended on the length of carbon chain and the best antibacterial performance was shown to be the 16-carbon chain QAS-modified chitosan. Furthermore, the morphology investigation of the TOB-loaded gelatine microspheres indicated the crosslinking degree, and the spherical shape could be effectively controlled by adjusting the amount of GTA. The relative long-term release of TOB from gelatine microspheres indicated its ability to cope with the slow healing of infected wounds. These data suggest that the polysaccharide-based encapsulating TOB-loaded gelatine microspheres could be used for bacteria-infected wounds. The in vitro and in vivo degradation of the polysaccharide-based hydrogel will be further studied and reported on in order to verify the possibility of its use in wound treatment.

## 4. Materials and Methods

### 4.1. Materials

Chitosan, Type B gelatine, Sodium Alginate (SA), and Calcium chloride (CaCl_2_) were all purchased from Aladdin Co. Ltd., Shanghai, China. DMEM (Gibco, Thermo Fisher Scientific, Inc., Waltham, MA, USA), MTT, and live/dead staining was purchased from Thermo Fisher.

### 4.2. Synthesis of Quaternary Ammonium Salts (QAS) with Different Carbon Chains

A series of quaternary ammonium salts with different carbon chains were synthesised. Briefly, 4.732 g epoxy chloropropane was added in a 50 mL round-bottom flask (4 replicates) and heated in an oil bath at 60 °C. Then, 2.811 mL of *N*,*N*-dimethyldecamine (Sigma, China); 2.732 mL of dodecyl dimethyl tertiary amine; 3.091 mL of tetacylethyl dimethyl tertiary amine; and 3.450 mL (molar ratio 4:1) of cetyldimethyl tertiary amine were added into the above four flasks, respectively. The flasks were sealed, and the reaction mixtures were stirred for 4 h. Subsequently, the reaction solutions were precipitated using hexane and centrifuged at 2000 r/min for 5 min. The precipitate was washed with hexane and ethanol under the same centrifugal conditions before vacuum-drying for 24 h. Finally, quaternary ammonium salts with different carbon chains were obtained and named as Ep-N +-C10, Ep-N +-C12, Ep-N +-C14, and Ep-N +-C16.

### 4.3. Characterisation of QAS

The structure of the quaternary ammonium salts was detected by nuclear magnetic resonance spectroscopy (^1^H-NMR). A total of 5 mg of a quaternary ammonium salt sample was dissolved into 600 µL deuterated chloroform reagent, and then added into the NMR tube for testing.

The epoxy value of QAS was determined by titration calorimetry. The titration process requires 20% tetraethylamine bromide glacial acetic acid solution and 0.1 mol/L perchloric acid glacial acetic acid solution (the standard solution concentration is recorded as C mol/L). The calibrated concentration of perchloric acid solution was 0.0947 mol/L. A total of 0.2 g of the quaternary ammonium salt dissolved in 10 mL of acetic acid was weighed and transferred to a 50 mL round-bottomed flask at room temperature on a magnetic stirring table. An additional 5 mL of acetic acid was used to transfer the quaternary ammonium salts as completely as possible. The solution changed from transparent to purple by adding 5 mL tetraethylamine bromide solution and dropping 0.1% crystal violet in volume fraction for 4–5 d. The perchloric acid solution was slowly added, and the amount of perchloric acid dropped V mL was recorded. The titration process changed from violet to blue and then to green, and the titration endpoint was considered when it turned to a stable green and never returned to blue. The epoxy value was calculated using the following equation:$$E=\frac{V\times C\times M/1000}{m}\times 100\%$$
where

*E* is the % epoxy value of quaternary ammonium salt sample;

*V* is the volume of perchloric acid standard solution consumed in the titration process;

*C* is the concentration of perchloric acid standard solution mol/L;

*m* is the mass of quaternary ammonium salt sample;

and *M* is the molecular weight of quaternary ammonium salt sample.

### 4.4. Microorganisms and Culture Media

Gram-negative *E. coli* (CMCC44103) and Gram-positive *S. aureus* (CMCC26003) were chosen to evaluate the antibacterial activity of the materials obtained in this study. *E. coli* was grown in LB Broth and *S. aureus* in TSB media.

### 4.5. Antibacterial Activity Test of the QAS

The antibacterial ability of the quaternary ammonium salts was tested via a minimal inhibitory concentration (MIC) assay. A pure single *E. coli* colony from the agar plate was inoculated into the LB broth and grown overnight in an incubator (37 °C). A stock solution of 1 mg/mL of tested compound (e.g., Eq-N_+_-C^10^) was prepared by dissolving the quaternary ammonium salts in the bacterial media. Compound stock solutions were serially diluted from 1 mg/mL to 1 μg/mL using bacterial media. Bacterial suspension was added to the diluted and tested compound to obtain a final concentration of 3 × 10^5^ CFU/mL. Bacteria suspension without quaternary ammonium salts served as the negative control, bacteria culture media without bacteria suspension as the control group. The samples were incubated at 37 °C for 12 h in a shaker and the optical density (OD) at 600 nm was measured using a microplate spectrophotometer. All tests were repeated three times. The antibacterial rate was calculated using the following equation:$$Antibacterialratio\%=\frac{\left(ODexperimental-ODnegative\right)}{\left(ODcontrol-ODnegative\right)}\times 100\%$$
where

OD_experimental_: OD value of the experimental group;

OD_positive_: OD value of the control group;

OD_negative_: OD value of the negative group.

The MIC of the compounds was determined by monitoring the pure growth of bacteria in nutrient solution with different polymer concentrations. MIC value is the concentration of the compound at which no bacteria growth was detected. The same method was used to analyse the MIC of Eq-N_+_-C^12^, Eq-N_+_-C^14^, and Eq-N_+_-C^16^ and the MIC of all the quaternary ammonium salts against *S. aureus*.

### 4.6. Preparation of QSA-Modified Chitosan (CMCS)

The synthesis of QSA-modified chitosan followed a previously reported method [29]. Briefly, 1 g of carboxymethyl chitosan (CMC) was added to 0.01 M NaOH solution in a 200 mL two-sided round-bottomed flask at room temperature and heated at 60 °C in an oil bath after it had completely dissolved. A certain amount of QAS with a carbon chain length of C10 was weighed and dissolved in 0.01 M NaOH solution before it was added to the round-bottom flask and stirred at 200 r/min for 24 h. The reaction solution was adjusted to slight acidity by adding 200 μL of acidic solution to obtain the crude product of QAS-modified chitosan (denoted as CMCS-N^+^). Then, the reaction solution was cooled at 4 °C and precipitated with 20 mL ethanol. The precipitation was centrifuged at 2200 r/min for 5 min and washed 2–3 times with ethanol. Finally, pure QAS-modified carboxymethyl chitosan with carbon chain length of C10 (recorded as CMCS-N_+_-C^10^) was obtained after vacuum drying in an oven for 12 h. QAS-modified carboxymethyl chitosan with carbon chain lengths C_12_ (CMCS-N^+^-C^12^), C_14_ (CMCS-N^+^-C^14^), and C_16_ (CMCS-N^+^-C^16^) were synthesized using the same method.

The structures of carboxymethyl chitosan and carboxymethyl chitosan quaternary ammonium salt were detected by ^1^H-NMR. The method of sample preparation is to add 5–8 mg of sample into 600 µL D_2_O reagent to fully dissolve for detection.

### 4.7. Antibacterial Activity Test of the CMCS

A similar method of antibacterial activity testing was used to determine the antibacterial performance of the chitosan quaternary ammonium salts. The concentration range of the tested samples was between 0.25 mg/mL and 16 mg/mL.

### 4.8. Preparation of Gelatine Drug Delivery Particles

Gelatine microspheres were prepared using the emulsification crosslinking method. Briefly, gelatine (0.5 g) was dissolved in 4 mL of ultra-pure water at 50 °C. TOB (0.05 g) was dissolved in 1 mL of ultra-pure water at room temperature and added dropwise to the dissolved gelatine solution. The mixed solution was then added to a 100 mL corn oil and preheated to 60 °C. A total of 0.5 mL of Span-80 was added into heated oil. The biphasic system was thoroughly mixed to form a without emulsion using a magnetic stirrer for 30 min. Then, the emulsion solution was chilled at 4 °C and 0.5 mL 25% glutaraldehyde was added to the cooling mixture. The mixture was stirred for 1 h before demulsification was carried out using isopropanol. Then, ethanol was used to wash the demulsified solution. The gelatine microspheres were obtained after being vacuum-dried at room temperature overnight. The same method was used to prepare TOB-loaded gelatine microspheres (TOB-G) using GTA of 0.3 mL, 0.15 mL, 0.10 mL, and 0.05 mL.

### 4.9. Characterisation of TOB-Loaded Gelatine Particles

The gelatine microspheres were spread onto a double-sided adhesive tape fixed to an aluminium platform. The microspheres were sprayed with gold. The morphology of the gelatine microspheres was observed using a scanning electron microscope (SEM) (AMETEK^®^ Quanta 3D FEG machine).

The TOB release profile was determined by placing gelatine microspheres in Transwell inserts and immersing the inserts in PBS buffer in a 24-well plate without shaking. A total of 2 mL of solution in the wells was collected at pre-determined times and 2 mL of fresh PBS buffer was added to maintain a constant volume. The content of TOB was determined by the o-phthalaldehyde (OPA) method. The concentration of the released TOB was measured with a UV–vis spectrophotometer at 333 nm.

### 4.10. Hydrogel Formation

A total of 20 mg/mL sodium alginate (SA) solution and 20 mg/mL CaCl_2_ solution was prepared using ultra-pure water, respectively. A total of 50 mg of TOB-loaded gelatine microspheres (0.10 mL GTA was used for the microspheres’ preparation) were weighed and dispersed in 4 mL ultra-pure water. Subsequently, the microsphere solution was mixed with 200 mg of CMCS and stirred until the mixture was completely dissolved. CMCS fabricated from different carbon chains were used to prepare a series of hydrogels (CMCS/SA/TOB-G). Then, 0.5 mL sodium alginate solution was added in a dropwise manner to the liquid obtained above and stirred to obtain a homogenous hydrogel precursor. Lastly, hydrogels were formed by adding 50 μL CaCl_2_ solution to the stirring mixture.

### 4.11. Rheological Testing of the Hydrogels

Rheological testing was performed on a Discovery HR-2 Rheometer (TA Instruments) with steel parallel-plate geometry (20 mm diameter). The storage (G′) and loss (G″) modulus were measured under a constant strain of 0.05 and frequency, ranging from 0.1 to 10^3^ rad/s at 25 °C. The tested volume of hydrogel was 350 μL.

### 4.12. Antibacterial Analysis of the Hydrogels

Then, the antibacterial activity of the CMCS-N^+^-C_16_/SA/TOB-G (0.1 mL GTA) hydrogel was evaluated using *E. coli* and *S. aureus*, as reported previously [33]. First, the bacteria were incubated overnight at 37 °C with shaking. The bacteria suspension was then diluted to a concentration of a 10^6^ colony-forming unit (CFU)/mL. Then, the selected hydrogel formulation, the TOB-G, and CMCS-N^+^-C_16_ were diluted to a series of concentrations through the agar media and added to the peri dishes. Subsequently, 100 μL diluted bacterial suspension were seeded onto the solid agar surface and incubated for 4 h at 37 °C. Then, the plates were washed by 2 mL PBS to resuspend the viable bacteria. Suspensions from each dish were cultured in new agar plates at 37 °C for 24 h and the number of bacterial colonies were counted and recorded. Bacteria with no treated material were used as the control group. The antibacterial ratio (%) was calculated by the following equation:$$Antibacterial\;ratio\;(\%)=\frac{Con-M}{Con}\times 100\%$$
where

*Con* represents the bacterial colony counts of control group and *M* represents the experimental groups.

### 4.13. Hydrogel Micromorphology

The freeze-dried hydrogel scaffold was mounted on stubs and coated with gold. The structure of the scaffold was observed using a SEM (AMETEK^®^ Quanta 3D FEG machine) at the required magnification.

### 4.14. Cytocompatibility of the Hydrogel

The in vitro cytotoxicity of the hydrogel was tested following the instruction of ISO 10,993 by MTT assay. Hydrogel extract was prepared by immersing 1 g hydrogel in 5 mL of full cell culture media (DMEM, 10% FBS, 1% P/S) for 24 h at 37 °C. L929 with a concentration of 8000 per well were seeded in a 96-well plate and allowed to attach overnight. The extraction was co-cultured with seeded fibroblasts for 24 h and 48 h at 37 °C. Cells cultured with no extraction served as a negative control and cells cultured with polyurethane were used as the positive control. An MTT assay was carried out following the manufacturer’s instructions. Briefly, MTT stock solution was prepared by diluting MTT in DMEM at a concentration of 1 mg/mL. The cell culture media prepared as mentioned above were carefully removed from the plates at the examination time. Then, 50 μL of 1 mg/mL MTT stock solution was added to each test well and the plate was further incubated for 2 h in the incubator. The MTT solution was gently discarded and 100 μL isopropanol was added to each well to dissolve the precipitated formazan. Then, absorbance was read using a microplate reader at a wavelength of 570 nm. The reduction of viability was calculated by the following equation:$$Viability\;(\%)=\frac{OD570\;nm\;of\;extraction}{OD570\;nm\;of\;blank}\times 100\%$$

Live/dead staining was also used to detect the cell morphology after co-culturing with the hydrogel’s extract at 24 h. Calcein AM (green) stained for live cells and ethidium homodimer-1 (red) for dead cells. Images were taken using a fluorescent microscope.

## Figures and Tables

**Figure 1 gels-09-00219-f001:**
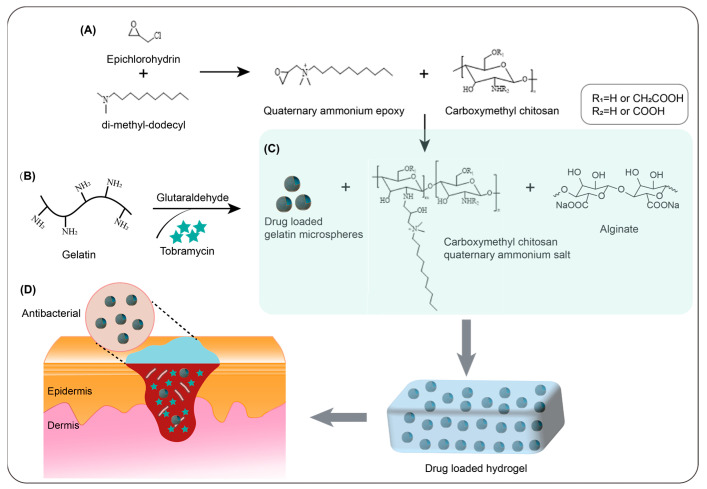
(**A**) Synthesis of quaternary ammonium salts. (**B**) Tobramycin-loaded gelatine microsphere formation; (**C**) Hydrogel formation. (**D**) Potential application of the hydrogel for wound-healing.

**Figure 2 gels-09-00219-f002:**
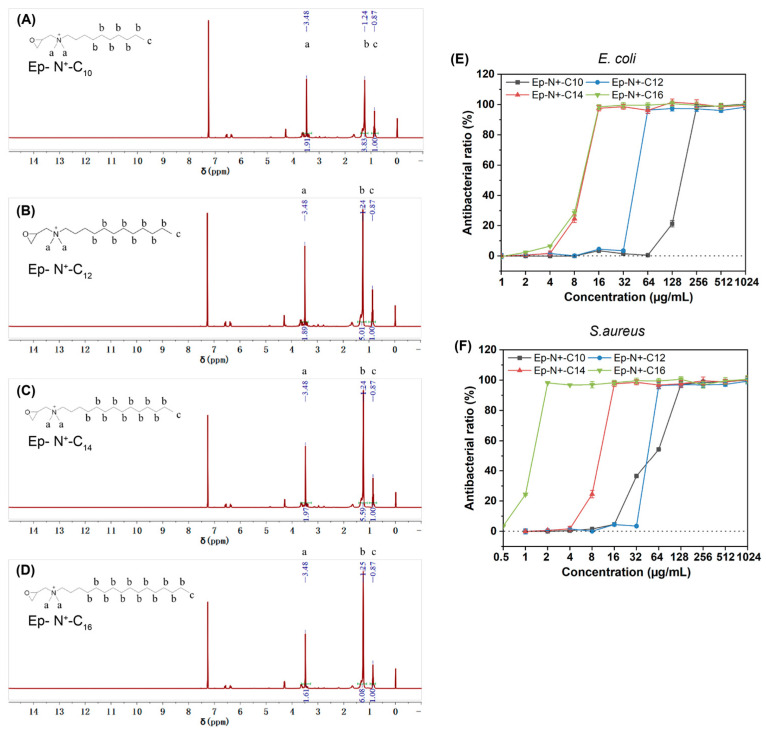
(**A**–**D**). 1H-NMR of synthesized QAS. (**E**,**F**). Antibacterial activity of QAS against *S. aureus* and *E. coli*.

**Figure 3 gels-09-00219-f003:**
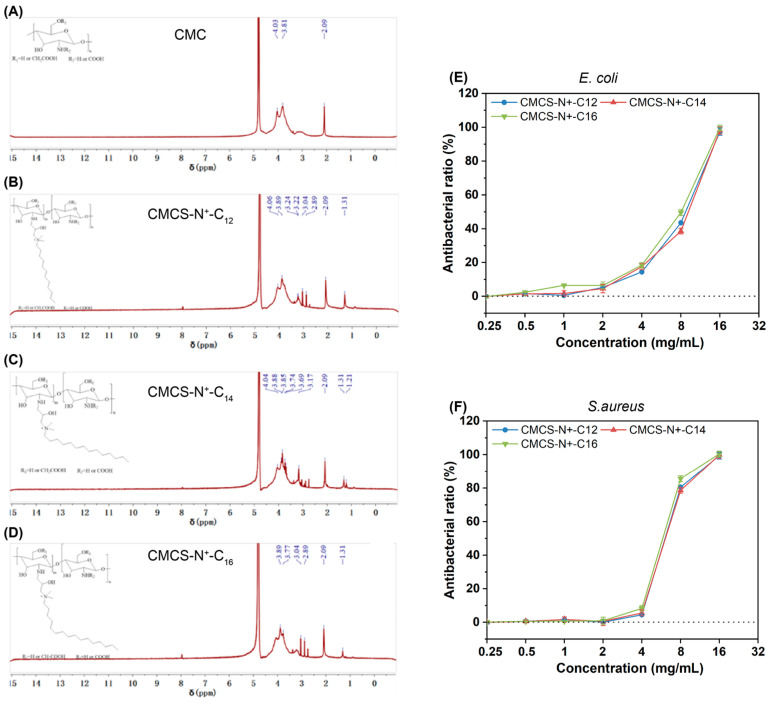
(**A**–**D**) ^1^H-NMR of synthesised QAS-modified chitosan (CMCS). (**E**,**F**) Antibacterial activity of CMCS against *S. aureus* and *E. coli*.

**Figure 4 gels-09-00219-f004:**
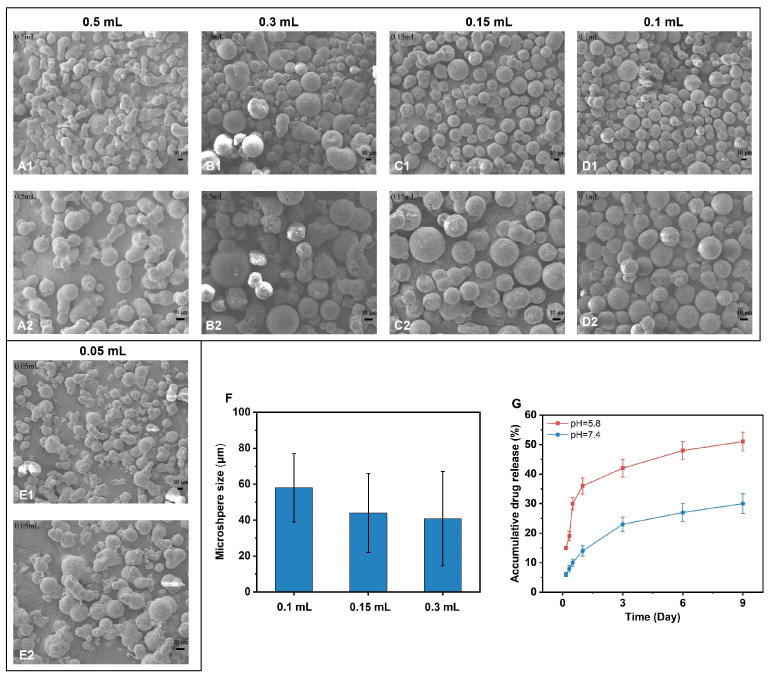
(**A**–**E**). Micromorphology of TOB-loaded gelatine microspheres formulated with different amounts of GTA. (**A**) 0.5 mL GTA; (**B**) 0.3 mL GTA; (**C**) 0.15 mL GTA; (**D**) 0.1 mL GTA; (**E**) 0.05 mL GTA. (**A2**–**E2**) are zoomed in images of (**A1**–**E1**). Scale bar, 10 μm. (**F**) The size of the microspheres formed by 0.3 mL, 0.15 mL, and 0.1 mL GTA. (**G**) Accumulative TOB release from the microsphere formulated with 0.1 mL GTA.

**Figure 5 gels-09-00219-f005:**
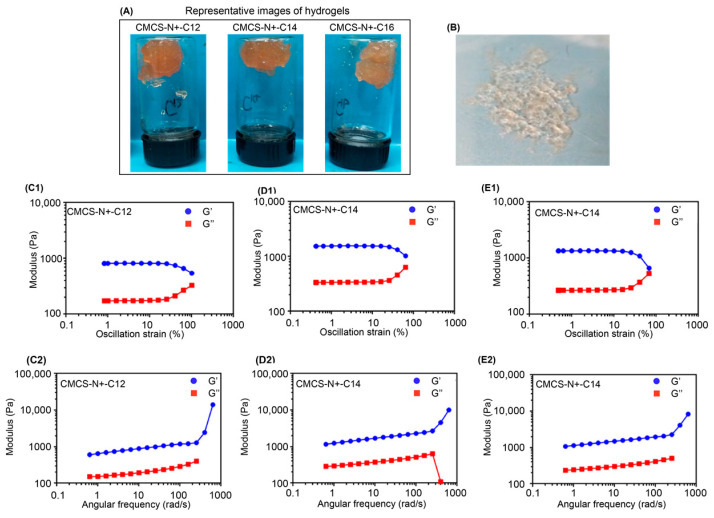
(**A**) Representative images of hydrogels formed by different CMCS. (**B**) Good performance of hydrogel spreading on the surface of Petri dish. (**C1**–**E2**) Rheological analysis of the formed hydrogels by oscillation strain and angular frequency at different conditions (CMCS-N^+^-C_12_, CMCS-N^+^-C_14_, and CMCS-N^+^-C_16_).

**Figure 6 gels-09-00219-f006:**
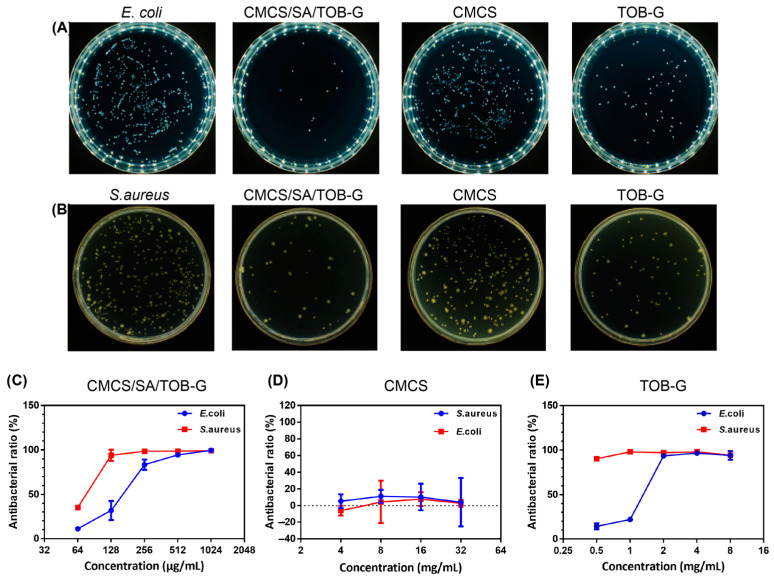
(**A**,**B**) Images of surviving bacterial colonies on agar plates. (**C**–**E**) Evaluation of the antibacterial activity of the materials.

**Figure 7 gels-09-00219-f007:**
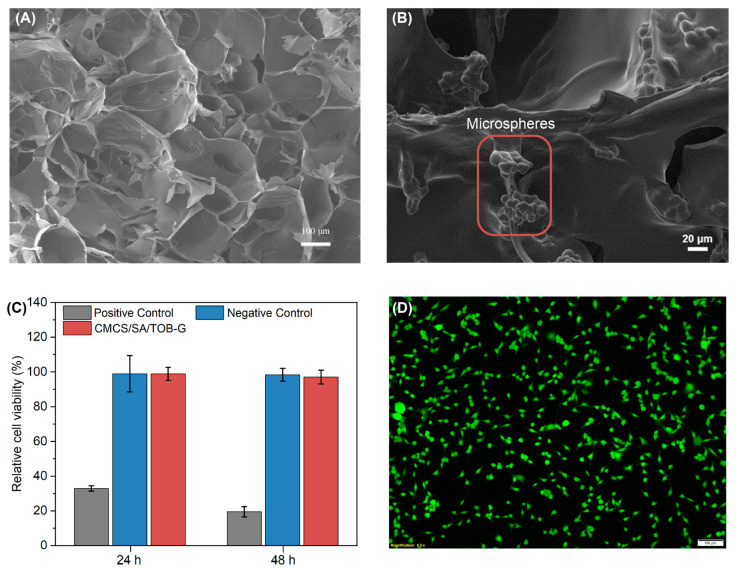
(**A**,**B**) Morphology of the hydrogel and the encapsulated microspheres using SEM. Scale bar, 100 μm and 20 μm. (**C**) MTT assay of the hydrogel. (**D**) Live/dead staining of the cells co-cultured with the hydrogel extract. Scale bar, 100 μm.

**Table 1 gels-09-00219-t001:** The epoxy values of QAS.

Sample	Molecular Weight (M) (g/mol)	Weigh (m) (g)	Perchloric Acid Standard Solution (V) (mL)	Epoxy Value (%)
C_10_-N+	277.5	0.2051	1.040	13.3254
C_12_-N+	305.5	0.1916	1.030	15.5526
C_14_-N+	333.5	0.1913	1.350	22.2877
C_16_-N+	361.5	0.2059	0.960	15.9615

**Table 2 gels-09-00219-t002:** The MIC of QAS with different carbon chain lengths for *E. coli* and *S. aureus*.

Sample	*E. coli* (MIC, μg/mL)	*S. aureus* (MIC, μg/mL)
Ep-N^+^-C_10_	256	128
Ep-N^+^-C_12_	64	32
Ep-N^+^-C_14_	16	32
Ep-N^+^-C_16_	16	2

## Data Availability

All data are available in the manuscript.

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
