# Peer review of "The Development of a Polysaccharide-Based Hydrogel Encapsulating Tobramycin-Loaded Gelatine Microspheres as an Antibacterial System"

_gels, 2023, doi:10.3390/gels9030219_

Round 1
Reviewer 1 Report
Figure 1d shows the microgels migrating into the wound which is not accurate nor supported by the data. It should show the drug, not the microparticles diffusing, or if it is in fact showing the tobramycin, then the sphere color and size should show that.
Figure 3 I find not useful, particularly parts a through e, as H1 NMR is really the wrong way to measure this and therefore it doesn't really support the author's claims. With polymers, solution proton NMR is not going to tell us much, and even just the presence of that very small CH2 peak doesn't mean it is attached to the polymer. I would have liked to see HPLC data to confirm, or MALDI. Also, for parts F and G of the same figure, why is it that the differences seen in figure 2 between the QAS disappear? I don't believe the charge neutralization proposed by the authors. Why include this in the paper if it isn't relevant?
Another point, glutaraldehyde is a toxic chemical and the crosslinking reaction is not quantitative. How do the authors propose to distinguish the MIC effects from any residual glutaraldehyde?
What is the mechanism behind the change in accumulative drug release as a function of pH? What is the physiological relevance of a pH of 5.8 vs 7.4? For Figure 4 I would expect to see a quantification of the particle sizes. What was the n?
Figure 5, the axis are way too small that it is incredibly hard to read, and impossible to read the legends.
For Figure 6, why is the antibacterial ratio? This is not a standard measurement and the authors don't describe what it actually is measuring. If the images are colonies formed, why not use that as the metric like everyone else? Also, from the so-called representative images, it doesn't look like the hydrogel killed "95% of S aureous.
Where is the Fickian data to show that microspheres are being released? What about the the release of drugs, or the activity of drugs?
Author Response
From Reviewer 1
- Figure 1d shows the microgels migrating into the wound which is not accurate nor supported by the data. It should show the drug, not the microparticles diffusing, or if it is in fact showing the tobramycin, then the sphere color and size should show that.
Re.
Thank you for your comments. In this figure we’d like to show the diffusion of microspheres from the hydrogel scaffold and the subsequent release of tobramycin. We have revised the Figure 1D based on your suggestion.
- Figure 3 I find not useful, particularly parts a through e, as H1 NMR is really the wrong way to measure this and therefore it doesn't really support the author's claims. With polymers, solution proton NMR is not going to tell us much, and even just the presence of that very small CH2 peak doesn't mean it is attached to the polymer. I would have liked to see HPLC data to confirm, or MALDI. Also, for parts F and G of the same figure, why is it that the differences seen in figure 2 between the QAS disappear? I don't believe the charge neutralization proposed by the authors. Why include this in the paper if it isn't relevant?
Re.
Thanks for your comments. We have revised the NMR spectrum. From the appearance of the new peak of ~1.3 ppm which is assigned to CH2 from the carbon chain. This is the clear evidence that the successful modification of chitosan derivatives. We have further revised in the manuscript. “By comparing the 1H-NMR results of carboxymethyl chitosan (Figure 3A) with CMCS, it can be seen that chemical shift δ = 1.3 ppm appears on the spectra after the addition of quaternary ammonium salts, this signal is assigned to the CH2 of the alkyl chains. (Figure 3B-3D).”
For the antibacterial effect of modified chitosan. We found that the antibacterial efficacy was weaker than the QAS. This is because free QAS are mainly used to change the permeability of the cells by aggregating on the surface of the bacterium, allowing water to enter, swell and rupture the bacterium. However, when QAS are conjugated to polymer chains, their ability to alter cell permeability is greatly reduced. Based on your comment, we have discussed this further in the manuscript. “In addition, free QAS are mainly used to alter cell permeability by aggregating on the surface of the bacterium, but when QAS are conjugated to polymer chains, their ability to alter cell permeability is greatly reduced.”
- Another point, glutaraldehyde is a toxic chemical and the crosslinking reaction is not quantitative. How do the authors propose to distinguish the MIC effects from any residual glutaraldehyde?
Re.
Thanks for your comments. Glutaraldehyde is a highly reactive crosslinking agent, and it is widely used to build different types of microspheres (Biomacromolecules 2011, 12, 9, 3186-3193, Polym. Bull. 2015, 72, 713-723). In our work, the microspheres are purified with ethanol. Therefore, there should be no residual glutaraldehyde and no effect on the MIC measurement.
- What is the mechanism behind the change in accumulative drug release as a function of pH? What is the physiological relevance of a pH of 5.8 vs 7.4? For Figure 4 I would expect to see a quantification of the particle sizes. What was the n?
Re.
We thank the reviewer’s comments. The pH sensitive character of the gelatin microspheres are due to the crosslinking of gelatin using glutaraldehyde. The structure of crosslink formed by Schiff base reaction of glutaraldehyde with amino and aldehyde groups is pH responsive. Relevant information and references have been added and highlighted in tracked changes in the Result and Discussion section in the manuscript.
The pH of a chronic wound exists in the range of 7.15-8.9. Researcher also reported the mean wound surface pH of the healing wounds was 6.91 compared with a mean pH value of 7.42 for the nonhealing wounds. (Influence of pH on wound-healing: a new perspective for wound-therapy. Arch Dermatol Res (2007)) Since infected wounds tend to proceed to nonhealing wounds, thus, we used pH value of 7.4 to mimic the pathological condition of a chronic wound and 5.8 of a normal tissue to perform the TOB release experiment.
The size of the TOB-loaded gelatine microspheres prepared by 0.1 ml GTA were presented in the Section 2.5. We have added the particle sizes of the rest microspheres to Figure 4G. The data was summarized by using an Image J software to calculate the mean particle size of each image. For each formulation, three images (n=3) were used to obtain the average particle size. And the revise Figure 4 is highlighted in the main text.
- Figure 5, the axis are way too small that it is incredibly hard to read, and impossible to read the legends.
Re.
We appreciate the reviewer’s pointing this out. Figure 5 has been revised and highlighted in the main text.
- For Figure 6, why is the antibacterial ratio? This is not a standard measurement and the authors don't describe what it actually is measuring. If the images are colonies formed, why not use that as the metric like everyone else? Also, from the so-called representative images, it doesn't look like the hydrogel killed "95% of S aureous.
Re.
We thank the reviewer’s comments on Figure 6. The antibacterial ratio reflects the percentage of bacteria colonies killed by the material compared with the control group. And this is also a widely used method to present the antibacterial performance. The counts of the colonies were generated by using Image J software. We have added high-resolution images of the agar plates which could be clearer to be identified. The same method can be found in the following published article,
Cryogenically printed flexible chitosan/bioglass scaffolds with stable and hierarchical porous structures for wound healing, Biomed. Mater. 2021, 16 015004.
- Where is the Fickian data to show that microspheres are being released? What about the the release of drugs, or the activity of drugs?
Re.
Thanks for your comments. In part 2.5, we have evaluated the release profile of TOB from microspheres. The results showed that TOB released faster in acidic environment. Moreover, in part 2.7. TOB loaded hydrogel showed better antibacterial activity compared to TOB-G and CMCS-N+-C16.
Reviewer 2 Report
From my perspective, this is an interesting work; however, the following issues have to be addressed carefully.
- In the abstract section, you need to focus more on quantitative information, not qualitative ones.
- The authors did not explain the novelty and significance of their work in the introduction part. Indeed, the introduction part is not cohesive. Topics change from sentence to sentence. The authors should follow the funnel procedure. The funnel technique for writing the introduction begins with generalities and gradually narrows your focus until you present your thesis.
- On page 2, line 51, the sentence "Hydrogel dressing can also function as a delivery platform, allowing it a promising alternative for hard-to-treat wounds, such as infected chronic wounds." needs also the following reference: https://doi.org/10.3390/ijms23073665
- In Results and Discussion, in each section, authors have reported and discussed many aspects, thus the key message from the section is not clearly understood. Authors are encouraged to reduce the texts by reporting the key findings, then having a discussion of the results with the relevant literature.
-The writing needs to be polished. The reviewer suggests rewriting the long sentences (especially the sentences in the Introduction and Conclusion parts) into short ones to make them easily understood. Authors should place the full stop or comma after the reference number in the bracket [ ], not before.
- Please unify the bibliographical references. Some references are separated by commas, others by hyphens. Please choose which one to use.
- The quality of all figures needs to be improved. Please remember that these figures will be shrunk once the paper is published and no one can read anything as a result.
- There are some formatting and spelling errors in this manuscript, a full-text check should be performed.
Author Response
From Reviewer 2
From my perspective, this is an interesting work; however, the following issues have to be addressed carefully.
We appreciate the reviewer’s comments on our work. The following issues have been addressed carefully.
- In the abstract section, you need to focus more on quantitative information, not qualitative ones.
Re.
We thank the reviewer’s suggestion on the abstract. The abstract has been revised and the revision has been highlighted.
- The authors did not explain the novelty and significance of their work in the introduction part. Indeed, the introduction part is not cohesive. Topics change from sentence to sentence. The authors should follow the funnel procedure. The funnel technique for writing the introduction begins with generalities and gradually narrows your focus until you present your thesis.
Re.
We appreciate the reviewer’s comments and suggestions in the introduction section. We have revised the introduction part as the reviewer suggested and highlighted in the main text.
- On page 2, line 51, the sentence "Hydrogel dressing can also function as a delivery platform, allowing it a promising alternative for hard-to-treat wounds, such as infected chronic wounds." needs also the following reference: https://doi.org/10.3390/ijms23073665
Re.
We thank the reviewer’s comments. More references have been added to the manuscript and highlighted in the main text.
- In Results and Discussion, in each section, authors have reported and discussed many aspects, thus the key message from the section is not clearly understood. Authors are encouraged to reduce the texts by reporting the key findings, then having a discussion of the results with the relevant literature.
Re.
We appreciate the reviewer’s suggestion on the manuscript. It helps us improve the quality of our manuscript. More detailed discussion has been added to the manuscript and highlighted in the main text.
- The writing needs to be polished. The reviewer suggests rewriting the long sentences (especially the sentences in the Introduction and Conclusion parts) into short ones to make them easily understood. Authors should place the full stop or comma after the reference number in the bracket [ ], not before.
Re.
We thank the reviewer’s comments on the manuscript. The English has been polished thoroughly and the layout of the cited reference has been revised.
- Please unify the bibliographical references. Some references are separated by commas, others by hyphens. Please choose which one to use.s
Re.
We thank the reviewer’s comments on this. The bibliographical references has been unified.
- The quality of all figures needs to be improved. Please remember that these figures will be shrunk once the paper is published and no one can read anything as a result.
Re.
We appreciate the reviewer’s pointing this out. The quality of all the figures have been improved.
- There are some formatting and spelling errors in this manuscript, a full-text check should be performed.
Re.
We thank the reviewer’s comments. We have checked and revised the full text.
Reviewer 3 Report
Zhang et al. describes the design of tobramycin loaded hydrogel composite system for anti-bacterial application. The concept of drug loaded microspheres incorporated in modified chitosan and sodium alginate based hydrogel is pretty simple and straightforward. The extend of drug release at physiological pH is quite interesting and the authors might add the mechanism of drug release or release kinetics in order to support their claim.
Author Response
Re.
We appreciate the reviewer’s comments on the research and manuscript. It really helps us improve the quality of the manuscript and our study. The discussion of the mechanism of the drug release at different pH has been added to the main text and highlighted.
Round 2
Reviewer 1 Report
Authors made minor revisions, when there were some major experimental design issues that were previously illustrated.
Reviewer 2 Report
Accept in the present form!